# Effect of core training on athletic and skill performance of basketball players: A systematic review

**Shengyao Luo[1,2], Kim Geok Soh[2], Yanmei Zhao** [3,4]*, **Kim Lam Soh[5], He Sun[2], Nasnoor Juzaily Mohd Nasiruddin[2], Xiuwen Zhai[6], Luhong Ma[2]**

**1** Faculty of Physical Education and Art, Jiangxi University of Science and Technology, Jiangxi Province, China, **2** Faculty of Educational Studies, Department of Sports Studies, Universiti Putra Malaysia, Serdang, Malaysia, **3** School of Foreign Languages, Yuxi Normal University, Yuxi, China, **4** Faculty of Educational Studies, Department of Foundation of Education, Universiti Putra Malaysia, Serdang, Malaysia, **5** Faculty of Medicine and Health Sciences, Department of Nursing, Universiti Putra Malaysia, Serdang, Malaysia, **6** Faculty of Educational Studies, Department of Language and Humanities Education, Universiti Putra Malaysia, Serdang, Malaysia

\* cherryxf99@yxnu.edu.cn

**Data Availability Statement:** All relevant data are within the paper and its Supporting Information files.

## Abstract

A limited number of studies focus on the effect of core training on basketball players' athletic performance and skills. This systematic review aimed to comprehensively and critically review the available studies in the literature that investigate the impact of core training on basketball players' physical and skill performance, and then offer valuable recommendations for both coaches and researchers. The data collection, selection, and analysis adhered to the PRISMA protocol. English databases, including Ebscohost, Scopus, PubMed, Web of Science, and Google Scholar, were searched until September 2022. A total of eight articles were included, with four studies comparing the effects of core training versus traditional strength training or usual basketball training. All studies investigated the impact of core training on athletic performance. The findings revealed that core training can help players improve their overall athletic and skill performance, particularly in the areas of strength, sprinting, jumping, balance, agility, shooting, dribbling, passing, rebounding, and stepping. In addition, core training, particularly on unstable surfaces, as well as combining static and dynamic core training, improve basketball players' athletic and skill performance. Despite the relatively little evidence demonstrating the effect of core training on endurance, flexibility, and defensive skills, this review demonstrates that it should be incorporated into basketball training sessions.

## Introduction

Basketball is a popular sport that necessitates technical, tactical, psychological, and physiological abilities [1]. Physical fitness, including speed, strength, endurance, agility, and flexibility, as well as jumping, running, balance, and direction shifting, all affect basketball performance

**Funding:** The authors received no specific funding for this work.

**Competing interests:** The authors have declared that no competing interests exist.

[2,3]. The physical demands of basketball can be evaluated in terms of physiological reactions such as elevated blood lactate concentration and sustained high heart rate [4], and physical activity indices such as total distance covered, distance covered at more than 18 km/h$^{-1}$(high-speed running), and the amount of high-intensity accelerations and decelerations [5]. However, due to age, it is hard for young basketball players to attain a speed greater than 18 km·h$^{-1}$ and a high-speed running distance.Due to improved decision-making and game interpretation, experienced players had lower physical demand values [6].

Traditional strength training has long been used to improve athletes'fitness. European and American experts began to widely implement strength training principles in a variety of sports in the late 1990s [7]. In this type of strength training, the training load gradually increasesduring training sessions [8]. This training separates the body's movement chain and disregards the strength training of the core muscles [7]. Therefore, this strategy has yielded the least obvious results, as players do not demonstrate functional carryover as a result of this type of training [9]. Subsequent analysis revealed that an unstable body state during movement prevented strength in the stable state. Therefore, it is challenging for strength to affect movement during gameplay [10]. This may explain why some athletes have outstanding strength during equipment-specific strength training but perform poorly on the field [10].

In contrast, the idea of core strength was derived from the study of core stability and appliedto human rehabilitation [11]. Core strength is a crucial requirement inmany sports. It not only ensures proper posture and facilitates daily tasks such as walking and climbing stairs but also serves as a brace and source of stability [12,13]. To complete technical motion, core training employsthe sports chain (power chain) principle, in which activities are linked in a "chain".So, each body part involved in the activity is linked. The completion of a technical activity depends on the transmission of momentum between each link, and the core force plays a "central" role in the momentum transfer process in the power chain [14]. Core training is therefore a novel approach to increasing strength transfer and coordinating muscle use and management during functional activities such as sport-specific skills. Sincethis sport involves the entire body, multiple muscle groups simultaneously participate in multiple dimensions [10]. The question then becomes "how does core training enhance athlete performance?"

Core training programsfocuson core stability and strength exercises [15]. Core strength refers to the ability of the muscles to generate force through contractile forces and intra-abdominal pressure. Core stability refers to the ability to support the spine due to muscle activation [16]. Core stability training employs static or slow motions and is primarily used in rehabilitation [15]. Core strength training, on the other hand,employs resistive and dynamic movements and is a realistic and safe way to improve health (i.e., flexibility and strength) and ability (i.e., coordination, balance, and speed) [13,15].

Athletes require more intensive core exercises due to the increased physical demands of competitive sports. Core stability training for low-load and motor control, which is an important component of core stability and core strength training, is often omitted from many training regimens [17]. By disregarding core stability training, athletes are unable to manage and utilize the entire body's muscle strength, increasingthe injury risk [16]. It is believed that high-load core strength training increases muscle strength, but low-load core stability training improves the central nervous system's ability to regulate muscle coordination and, hence, movement efficiency [17]. Therefore, athletes' core training programs should include both low-load core stability training and high-load core strength training [15].

In addition, new exercises using unstable surfaces, such as Swiss balls, have been developed to increase the proprioceptive demands of exercises [18]. A previous study found that practicing core exercises on unstable surfaces alters muscle activity and the way the muscles work together to stabilize the spine and the entirebody [19]. Therefore, exercise on unstable surfaces

necessitates a motion control system to stabilize the muscles surrounding the spine, increase core muscle activity, and improve muscle recruitment mode [20,21]. Furthermore, it promotes neuronal adaptation, neuromuscular recruitment, effective motor unit synchronization, and effective proprioceptive feedback while lowering neuroinhibitory reflexes [22,23]. As a result, deeper muscle groups may be encouraged to engage in movement [24]. Ultimately, this adaptation enhances the body's stability during movement and supports technical movements both at rest and in motion [25]. All of the aforementioned processes are crucial to therapeutic or athletic training because they contribute to proper movement execution [15].

There is a debate about the relationship between motor abilities; one viewpoint holds that motor abilities are highly related to one another (the general motor abilities hypothesis), while the opposing view holds that they are relatively independent of one another (the specific motor abilities theories) [26]. However, understanding the various points of view will aid in applying the concept of motor abilities to motor skill performance achievement [27]. The general motor abilities hypothesis has been around since the early twentieth century [28,29]. It is assumed that if a person is highly skilled in one motor skill, he or she will be or will become highly skilled in all motor skills. This prediction is based on the fact that there is only one general motor ability [26,27]. The specificity of the motor abilities hypothesis is an alternative viewpoint that has received widespread support. Franklin Henry is widely credited with developing the specificity hypothesis in order to explain research findings that the general motor ability hypothesis could not explain [30]. According to this specificity viewpoint, individuals have numerous motor abilities that are relatively independent. This means that, for example, if a person demonstrated a high level of balancing ability, we couldn't predict how well that person would perform on a reaction time test [26]. In sum, the specificity hypothesis is likely helpful in understanding in what ways core strength training may contribute to improved sport-specific performance.

In recent years, core training has been shown to improve core stability but not performance with low specificity or functional carryover. For instance, core trainingimproved men rowers' core endurance but did not improve their functional performance in the vertical jump, shuttle run, or 40-meter sprint [31]. Another study discovered that core trainingimproved core stability but had no effect on physical performance, such as the myoelectric activity of the abdominal and back muscles, running performance (treadmill VO2max), running economy, or running posture [32]. Although numerous media outlets have described the efficacy of "core training,"leading athletes to believe it will improve their competitive performance, the scientific community remains uncertain about the relationship between core training and athletic performance [33]. This systematic review aims to elucidate the impact of core training as a function of the above-noted specificity hypothesis on basketball players' physical and skill performance.

## Methods

### Search strategy

The data collection, selection, and analysis adhered to the Preferred Reporting Items for Systematic Reviews and Meta-Analyses (PRISMA), and this review was registered on the INPLASY website (https://inplasy.com/), with registration number INPLASY2021100013 and DOI number 10.37766/inplasy2021.10.0013 [34]. In this study, English databases such as Ebscohost, Scopus, PubMed, Web of Science, and Google Scholar were searched untilSeptember 2022. Each search was conducted by title/abstract, with the following terms serving as the primary retrieval criteria: ("Core Strength Training" OR "Core-Strength Exercise" OR "Core-training" OR "Core-stability Exercise" OR "Core-stability Training") AND ("Athletic

Performance" OR "Physical Performance" OR "Basketball Skills" OR "Offensive Skill" OR "Defensive Skill").

## Eligibility criteria

In this review, PICOS (population, intervention, comparison, and outcome)wereused as inclusion criteria to conduct a literature search [35].The inclusion criteria were as follows:

1. Experimental studies on core training and basketball players' athletic and skill performance.

2. A population of healthy basketball players of all ages and sexes.

3. At least four weeks of core training on either stable or unstable surfaces.

4. Anoutcomeaddressing the impact of at least one core strength intervention onbasketball players' athletic or skill performance.

   The exclusion criteria wereas follows:

1. Articles with no full text.

2. Articles published in languages other than English or Chinese.

3. Review articles, conference papers, book chapters, or unpublished articles.

4. Studies with no intervention or in which core training was not the primary intervention.

5. Studies on players from other sports rather than basketball.

## Study selection

The articles had to meet the inclusion criteria and were chosen and included independently by two authors. After removing duplicates, one author reviewed the titles and abstracts to decide which papers should be included in this study. In the event of a disagreement between the two authors during the selection of a single article, a third author was consulted to review the entire paper and make a decision on its inclusion.

## Data extraction

The following data were extracted from the included studies: (1) population characteristics (type, number, sex, and age); (2) intervention (type, main exercise, training arrangement, duration, and frequency); and (3) main outcome.

## Quality assessment

The quality of the included studies was assessed using the Physiotherapy Evidence Database (PEDro) scale (www.pedro.org.au). It has excellent validity and reliability for assessing experiment method quality. Each manuscript was evaluated using 11 criteria,each of which wasassigned a score of 0 or 1. Therefore, the total PEDro score of each study ranged from 0 to 10, with higher values indicating a higher degree of methodological rigor.

## Results

### Search results

Fig 1 depicts the literature review process. The initial search yielded232 articles,146 of which remained after duplicates were removedusing Endnote software.Furthermore, 60 articles were

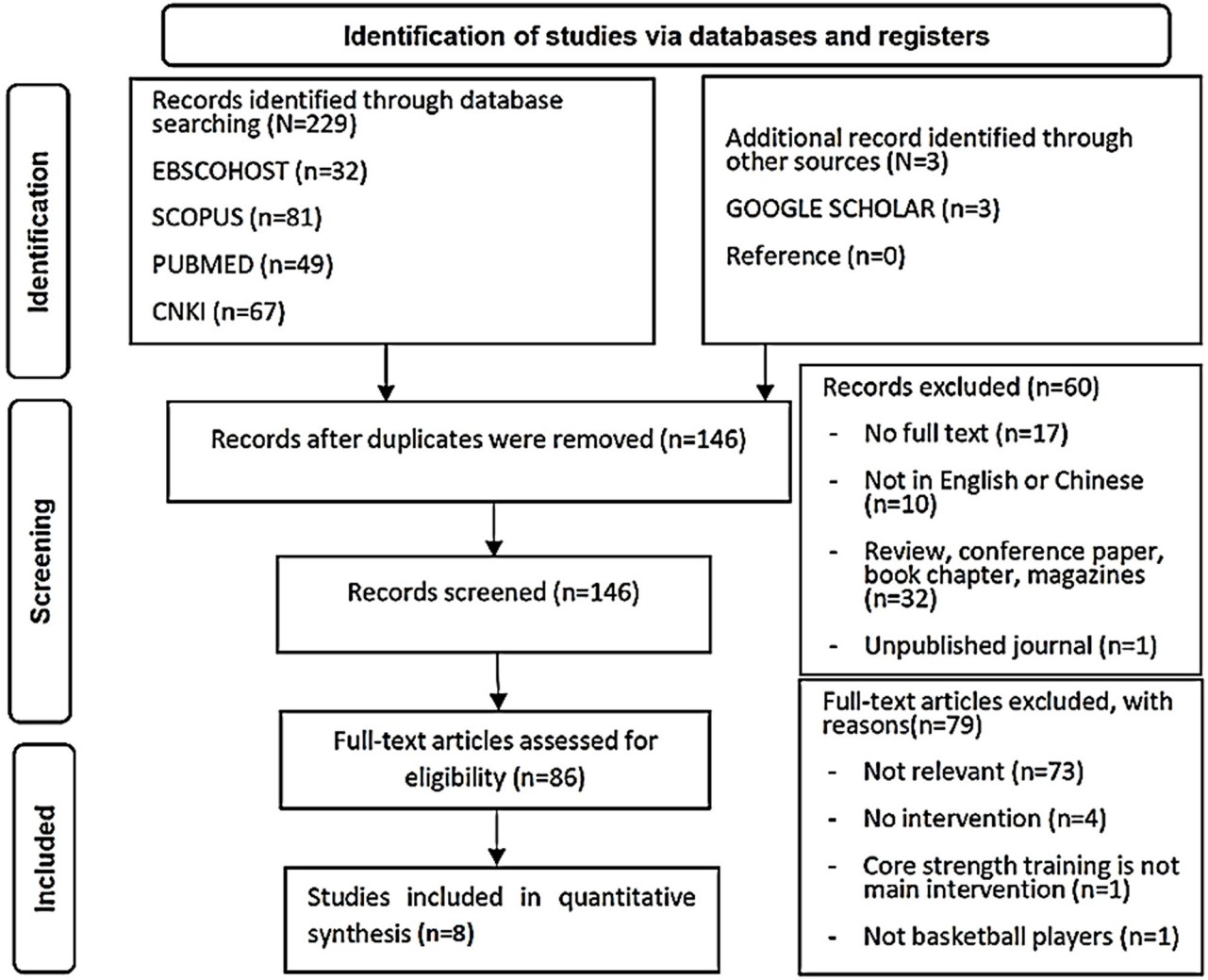

**Fig 1. The PRISMA flow chart for the search, screening, and selection strategy for the eligible studies.**

excluded (17 articles with no fulltext, 10 not written in English or Chinese, 32 not research articles, and one unpublished). The remaining 86 full-text articles were assessed for eligibility, and78 were excluded (73articles with unrelated topics, fourwith no intervention, onethat did not involve core training, and one that did not involve basketball players). Finally, eight articles were included in the quantitative synthesis.

## Participant characteristics and study design

Theparticipant characteristics of the eight included studiesare summarized in Table 1. Only three studies included university students as participants [36,37], while the other studies did not report the characteristics of the participants. Except for two studies that did not report sex [38,39], the majority of participants were males($n$ = 129),with only 35 females. Furthermore, all studies provided detailed information on the effects of core training on participants. However, only four studies compared the effects of core training on basketball skills versus traditional strength training or usual basketball training [36,38–40]. All studies investigated the impact of core training on athletic performance [36,41].

**Table 1. Populations, interventions, and main outcomes of the included studies.**

| Study | Population Characteristics (Type/number/sex/age) | Interventions | | | | Main Outcome |
|---|---|---|---|---|---|---|
| | | Type | Main Exercise | Training Arrangement | Duration & Frequency | |
| Li, 2022 [36] | 12 university students M EG: 20.5 ± 1.33 years CG: 20.8 ± 0.86 | EG: Core strength training CG: Traditional training | Superman plank, Sit-ups, Side bridge, Swiss ball exercises | N/A | 12weeks | Dribble layup↑, Shooting skills ↑, Run-up ↔ |
| Ning,2020 [38] | 30 college students N/A EG: 18.4 ± 0.67 years CG: 18.5 ± 0.59 years | EG: Core strength training CG: Traditional training | Supine buttocks, Leg lift | N/A | 10 weeks | Lower limb strength↑, Approach height↑, Vertical jump ↑, Number of consecutive jumps in 15 seconds↔, Jump shot ↑, T-test ↑ |
| Yilmaz, 2022 [39] | 16 basketball players N/A 13.29 ± 1.96 years | EG: Isometric core strength training CG: Usual basketball training | Plank (prone, supine, right and left side) | 30 sec × 2 reps | 4weeks 2 times/week 12 min/session | Modified T agility test↑, Sprint ↔, Vertical jump ↔ |
| Dogan and Savas,2021 [40] | 30 basketball players M 12–14 years | EG: Core strength training CG: Usual basketball training | Plank (prone, right and left side, superman), Squat, Russian twist, Medicine ball exercises | Week (1–2) 20 sec × 3 sets Week (2–4) 20 sec × 4 sets Week (5–6) 20 sec × 5 sets Week (7–8) 20 sec × 6 sets | 8weeks 3 times/week 45–60 min | Core muscle strength ↑, Stabilometer Balance ↑, Y-Balance ↑, Dribbling with Right Hand↑, Chest Pass ↑, V-Cut↑, Taking Pass↔, Jab-Step↑, Dribbling and Right Layup↔, Rebound↑, Overhead Pass↑, Cut Towards Left ↑,Taking Pass and Dribbling with Left Hand ↑, Dribbling with Left/right Hand Between The Legs ↑, Cross Over Dribbling (Left/right)↑, Left Lay Up↑, Cross Over Dribbling Behind the Back with Left Hand ↑, Reverse Dribbling with Right Hand↑, Hesitation (Dribbling)↑, Jump Shot↑ |
| Sannicandro, 2020 [42] | 42 basketball players 16 F/26 M 8.22 ± 0.40 years | EG: Core training CG: Routine training | Plank, Plank with hand on unstable tools | 3 ×10 rip × 3 sec 2 ×8 rip × 3 sec × side 3 ×8 rip × side 4 × 6 rip × limb | 4weeks 2 times/week 1 hour/session | Jump ability ↑, 10 meter sprint ability↑ |
| Sannicandro, 2017 [41] | 44 basketball players 19 F/25 M 7.07 ± 0.30 years | EG: Core stability training CG: Routine training | Plank on the ground, Plank with hands on unstable tools | 2 ×8 rip × 3 sec 2 ×8 rip × side × 3 sec 2 ×8 rip × 3 sec 3 × 6 rip × limb | 4weeks 2 times /week 1 hour/session | Jump ability ↑ |
| Şahiner and Koca, 2021 [43] | 22 basketball players M 16–18 years | EG: Core training CG: Routine training | Superman, Cat and Camel Toe touch, Front plank | 3 sets ×45 sec × 45s recovery 3 sets ×45 sec × 45s recovery 3 sets ×45 sec × 45s recovery 3 sets ×45 sec × 45s recovery | 8 weeks 2 times/week 30–60 minutes/session | Free-throw ↑, vertical jump ↑ |

*(Continued)*

**Table 1.** (Continued)

| Study | Population Characteristics (Type/number/sex/age) | Interventions | | | | Main Outcome |
|---|---|---|---|---|---|---|
| | | Type | Main Exercise | Training Arrangement | Duration & Frequency | |
| Hu and Li, 2019 [37] | 14 university male basketball players M 21.50 ± 1.09 years | EG: Core strength training | Superman plank, Plank, Sit up, Sit on the balance pad and turn the body around | N/A | 12weeks 3 times/week 1.5hours/ session | Lower limb, balance control ability↑ |

M, male; F, female; N/A: No report; CG, control group; EG, experimental group; ↑, significant within-group change from pretest to post-test; ↔, non-significant within-group change from pretest to post-test.

## Quality assessment using the PEDro scale

A comprehensive evaluation of each study using the PEDro scale is presented in Table 2. All studies were rated 3 to 6 on the PEDro scale.

## Training programs

Table 1 summarizes the training characteristics of basketball players whoemployed core training as an intervention [36,41]. The intervention included core strength training [36,37,40], core training [42,43], isometric core strength training [39], and core stability training [41]. Meanwhile, only two studies employed core training onunstable surfaces [41,42]. The findings revealed that core training on unstable surfaces was associated with improved jump [41,42] and 10-meter sprint abilities [42].

In terms of duration and frequency, the traininglasted four weeks [39,41,42], eight weeks [40,43], 10 weeks [38], or12 weeks [36,37]. The four-week training enhanced agility [39], jump ability [41,42], and 10-meter sprint ability [42]. However, training for more than four weeks improved skills [36], core muscle strength [40], and balance control ability [37,40]. In addition,

**Table 2. Summary of methodological quality assessment scores of all included studies using the PEDro scale.**

| Reference | Li, 2022 [36] | Ning, 2022 [38] | Yılmaz, 2022 [39] | Dogan and Savas, 2021 [40] | Sannicandro, 2020 [42] | Sannicandro, 2017 [41] | Şahiner and Koca, 2021 [43] | Hu and Li, 2019 [37] |
|---|---|---|---|---|---|---|---|---|
| Eligibility criteria | 0 | 0 | 1 | 0 | 0 | 0 | 0 | 1 |
| Random allocation | 1 | 0 | 0 | 0 | 1 | 0 | 0 | 0 |
| Allocation concealment | 0 | 0 | 0 | 0 | 0 | 0 | 0 | 0 |
| Group similar at baseline | 1 | 1 | 1 | 1 | 1 | 1 | 1 | 1 |
| Subjects blinding | 1 | 0 | 0 | 0 | 0 | 0 | 0 | 0 |
| Therapist Blinding | 0 | 0 | 0 | 0 | 0 | 0 | 0 | 0 |
| Assessor blinding | 0 | 0 | 0 | 0 | 0 | 0 | 0 | 0 |
| Less than 15% dropouts | 1 | 1 | 1 | 1 | 1 | 1 | 1 | 1 |
| Intention to treat analysis | 0 | 0 | 0 | 0 | 0 | 0 | 0 | 0 |
| Between group comparisons | 1 | 1 | 1 | 1 | 1 | 1 | 1 | 0 |
| Point measure and variability | 1 | 1 | 1 | 1 | 1 | 1 | 1 | 1 |
| PEDro scale | 6 | 4 | 4 | 4 | 5 | 4 | 4 | 3 |

training wasperformed twice a week [39,41,42] orthree times a week [37,40]. However, two studies did not report its training frequency [36,38]. Training twice a week improved agility [39], jump ability [41,42], and 10-meter sprint ability [42], whereas training three times a week improved core muscle strength [40]and balance control ability [37,40].

In addition, most studies' training sessions were about 30–60 minutes and resulted in better jumping abilities [40,41]. However, the shortest training session was 12 minutes and improved agility [39], while the longest training session was 1.5 hours and improved balance control ability [37]. However, two studies did not report the training length [36,38].

In terms of the main exercise, the plank exercises were involved in nearly all studies, except one [38]. Supine buttocks, leg lift [38], Swiss ball exercises [36], medicine ball exercises [40], sit-ups [36,37], cat and camel [43], toe touch [43], and Russian twist [37,40] werealso involved in the core training program. Five studies provided a more thorough description of their core training programs [39–43]. A study showed that all exercises, including plank (prone, supine, right and left side), were performedfor 30 seconds with two repetitions in each training session [39]. Another study usedplank (prone, right and left side, Superman), squat, Russian twist, and medicine ball exercises for 20 seconds with three sets. In particular, the number of training sets was increased by one every two weeks [40]. Plank, side plank, and Superman are the most prevalent exercises in theresearch, with 2–3 sets of 8–10 repetitions of plank and side plank exercises commonly advised [41,42]. Moreover, mountain climbers wereinvolved in 3–4 sets [41,42]. A study demonstrated that core training exercises can be performed with the same number of sets, repetitions, volume, and recovery period. Superman, cat and camel, toe touch, and front plank may be performed after three sets of 45 seconds with 45 seconds of rest [43].

## Outcomes

**3.5.1 Effect of core training on strength.**  Strength helpsathletes perform better [16,44]. Two studies examined strength [38,40]. Onestudyon 30 basketball players [40] revealed a significant difference in post-test core muscle strengthbetween the experiment and control groups (122.44 ± 20.18 vs. 91.10 ± 38.11, respectively, $p < 0.05$). However, another study involving 30 college students found a significant difference in lower limb strength after intervention between the experiment and control groups (182.3±23.7 vs. 175.7±25.8, respectively, $p < 0.05$) [38].

**3.5.2 Effect of core training on sprint.**  Two studies examined core training's effect on sprinting. A study on 42 participants (8.22 ± 0.04 years, 16 females and 26 males) comparingcore and routine training revealed a change in training (core strength) group test scores between pre-and post-interventionin the 10-meter sprint (4.65 ± 1.09 sec vs. 3.89 ± 1.11 sec, respectively, $p < 0.005$) and in the $10 \times 5$-meter test (33.66 ± 3.15 sec vs. 31.51 ± 3.86 sec, respectively, $p < 0.001$). The $10 \times 5$-meter test results showed a significant difference between the control (regular training) and training groups (34.69 ± 2.98 sec vs. 31.51 ± 3.81 sec, $p < 0.05$) [42]. In contrast, another study on 16 basketball playersshowed no significant difference in sprint test results between the experiment and control groups after intervention ($p > 0.05$) [39].

**3.5.3 Effect of core training on jump.**  The effect of core training on jumping ability was examined in five studies [36,38,41,42]. To assess jump performance, a study involved three tests, including approach height, vertical jump, and the number of consecutive jumps in 15 seconds,revealed a significant difference between the experiment and control groups in the post-test approach height (3.19±0.08 vs. 3.24±0.05, respectively, $p < 0.05$) and vertical jump performance (3.13±0.31 vs. 3.09±0.39, respectively,$p < 0.05$) [38]. Another study revealed a significant difference between the core training and control groups in the left limb side hop

(39.93 ± 4.69 cm vs. 40.98 ± 5.71 cm, respectively, $p<0.05$), right limb side hop (39.20 ± 5.90 cm vs. 41.92 ± 6.90 cm, respectively, $p<0.05$), left limb 6-meter timed hop (5.58 ± 0.45 sec vs. 4.28 ± 0.96 sec, respectively, $p<0.01$),and right limb 6-meter timed hop (5.58 ± 0.45 sec vs. 4.28 ± 0.96 sec, respectively, $p<0.01$) [42], indicating that the hopping and leaping skills of both legs improved following core training regimens.

On the other hand, a study found that the experimental group demonstrated a significant difference between pre-and post-core training testsin terms of left limb side hop (36.55 ± 6.32 cm vs. 38.98 ± 5.71 cm, respectively, $p<0.05$), right limb side hop (37.21 ± 6.51 cm vs. 39.27 ± 5.74 cm, respectively, $p = 0.001$), left limb 6-meter timed hop (5.82 ± 0.87 sec vs. 4.44 ± 0.94 sec, respectively, $p<0.001$), and right limb 6-meter timed hop(5.78 ± 0.77 cmvs. 4.42 ± 0.88 sec, respectively, $p<0.001$) [41]. Another study found a statistically significant difference between the experimental group's pre-test and post-test vertical jump scores (46.64 ± 6.61 vs. 52.00 ± 6.34, respectively, $p<0.05$) andwhen comparing the experimental group's vertical jump performance on post-tests to that of the control group (52.00 ± 6.34 vs. 46.82 ± 5.19, respectively, $p<0.05$) [43]. In contrast, there was no statistically significant difference between the experimental and control groups in post-test run-up scores ($p = 0.917$) [36].

**3.5.4 Effect of core training on balance.**   The effect of core training on balance was investigated in two studies. The first one employed four different standing postures [37]. The center of gravity of the movement area and the center of gravity velocity of the left and right foot were recorded with the eyes open and closed, and there was a significant difference between the pre-and post-training time periods of participants (312.50 ± 62.54 vs. 198.43 ± 57.91; 2.30 ± 0.44 vs. 2.06 ± 0.44; 202.00 ± 70.31 vs. 125.86 ± 39.48; 2.06 ± 0.37 vs. 1.73 ± 0.33; respectively, $p<0.001$) with eye openand (2439.36 ± 713.72 vs. 1848.07 ± 579.57, 6.70 ± 1.18 vs. 4.71 ± 1.02, 2027.00 ± 750.36 vs. 1468.57 ± 503.71, 6.32 ± 1.14 vs. 3.94 ± 1.08, respectively, $p<0.01$) with eye closed [37]. The second study examined the effect of core training on balance using the Stabilometer Balance test and the Y-Balance and found a significant difference in post-test results between the experimental and control groups ($p<0.01$) [40].

**3.5.5 Effect of core training on agility.**   The effect of core training on agility performance was examined in two studies [38,39]. TheT-test was used in one study [38], and it revealed a significant difference between the experiment and control groups after intervention (8.29 ±0.96 vs. 9.05±1.23, respectively, $p<0.05$) [38]. In addition, another study used the modified agility T-test andfound a significant difference between the control and experiment groups in the post-test after intervention (14.29±0.75 vs. 12.81±1.75, respectively, $p<0.05$) [39].

**3.5.6 Effect of core training on skills.**   The effect of core training on basketball skills was examined in four studies [36,38,40,43].A study compared the speed of the dribble layup and shooting test after intervention between the control and experimental groups and foundsignificant differences(36.8 ± 1.78 sec vs. 34.80 ± 1.89 sec, $p<0.05$; 8.3 ± 1.42 vs. 8.90 ± 1.78, $p<0.001$; respectively) [36]. In addition, another studyrevealed a significant difference between the experimental and control groups after intervention in jump shot percentage (79.8±3.6 vs. 59.2±3.4, respectively, $p<0.05$) and hit rate against jump shots (51.3±3.1 vs. 38.2±4.8, respectively,$p<0.05$) [38].

On the other hand, a study examined the effect of core training on 20 different basketball skill tests, including dribbling, passing, shooting, rebounding, and step tests. Except for the passing,dribbling, and right layup tests, all tests showed a significant difference between the control and experiment groups in the post-test ($p<0.05$) [40].Furthermore, another study involving free-throw tests revealed that the experimental group showed a significant difference between the pre-and post-tests (5.81±0.968 vs. 6.81±0.750, respectively, $p<0.05$), with no significant difference between the experimental and control groups in the post-test (6.81±0.75 vs. 6.93±0.83, respectively, $p<0.05$) [43].

## Discussion

There is a lack of consensus on whether an athlete should use strength training within a sport-specific training program, what type of strength training the athlete should use, when to use, and how to apply it [7,45,46]. General strength training can provide overall protective strength on the major muscle groups and can be beneficial during the off-season and even during the early base phase of training [47]. It increases overall protective strength and will help protect specific joints from stresses that can occur in everyday life as well as during specific training sessions. Adding additional muscle can also help with energy transfer, as a mix of fast and slow twitch fibers can provide more energy uptake, which can help athletes with endurance [47]. To build any type of muscle, however, the muscle must be overloaded, which causes fatigue and necessitates recovery. Furthermore, most coaches, trainers, and sport scientists understand that most athletes do not possess unlimited hours each day to train nor do they have limitless capacity to recover from training and competition, so they must be highly strategic in what they include in training plans [48,49].

As a result, highly specific strength training appears to be most beneficial to athletes, as it can improve the execution of sports activities, particularly as external resistance may be added to make the training task more difficult. Specific strength training can stimulate focused muscle growth, increasing further power, endurance, and speed of movement without requiring the excessive time commitments or undue fatigue that generalized strength training programs may be likely to generate [50]. This systematic review aimed to assess the impact of a specific strength training, i.e., core training, on basketball players' athletic and skill performance, as well as to offer recommendations for basketball coaches and potential directions for future research.

### Effect of core training on strength

For optimal basketball performance, all strength parameters must be boosted [51]. Core training is a dynamic proprioceptive training of core stabilizing muscles that can improve core control [14]. Some studies claim that core training can increase core muscle activation and strengthen trunk and hip muscles [52,53]. Unstable surface training, in particular, had a greater effect [52,53]. Meanwhile, several studies have confirmed that the core is the main anatomical and functional center from which all movements originate and are transmitted to the extremities [54,55]. Therefore, a stronger core could enhance limb strength performance.

### Effect of core training on sprint

Sprinting is important in basketball. Unfortunately, only two studies examined the sprint speed of young basketball players [39,42]. In one study, core training improved 10-meter sprinting ability [42]. Core strength training improved the stability of the core parts of the body, such as the spine and pelvis, as well as the stability and fluctuation of the center of gravity during fast running. Increased hip stability and flexibility improved athletes' range of motion, stride length, and stride frequency during the moving process [56]. Furthermore, strong core muscles play a crucial role in stabilizing and transferring lower limb energy [57]. During squatting and running, the core muscles absorb more energy. This condition improves muscle control, upper and lower limb coordination efficiency, energy consumption, and sprint performance [58].

### Effect of core training on jumping

Jumping is an important basketball skill that influences shooting and rebounding [59]. Five articles examined the jumping ability of basketball players of different ages [36,38,41,42].

However, four studies found that core training could improve jump ability [38,41,42]. This is due to the fact that core region muscles play an important role in transferring power to the extremities [60]. Meanwhile, increasing core strength improves pelvic, spine, and hip joint stability and provides a stable fulcrum for limb movement,resulting in more coordinated movements and better control of body stability during rapid movement changes [61]. Therefore, the participants demonstrated hip, knee, and ankle flexion, resulting in an overall stable body. Meanwhile, power transmission between the upper limbs, trunk, and lower limbs will be more coordinated [2].

## Effect of core training on balance

According to two studies, core training improves basketball players' balancing ability [37]. Balance is regulated by the vestibular apparatus within the central nervous system as well as receptors in muscles, tendons, and joints and then coordinated within the somatosensory cortex with the aid of visual stimulus [15]. Core training controls spine and pelvic stabilityina timely manner, coordinates shifting centers of gravity and posture adjustment during movement, and increases core stability, which improvesoverall balance ability [37]. In addition, core training stresses the fixation of stable and small muscles, allowing for the complete exercise of tiny muscle groups and nerve modulation. All of these effects promote proprioception and balancing abilities [62].

## Effect of core training on agility

Agility is defined as the ability to rapidly change one's body direction and position [63]. Core training was found to improve agility in two studies [38,39]. The core could be considered the center of the kinetic chain in sports activities. Strong core muscles improve motor recruitment, neural recruitment, and neural adaptation [64]. Therefore, increasing core strength and stability could be expectedto improve athletes' motor skills such as coordination, agility, speed, and movement balance [64].

## Effect of core training on basketball skills

Four studies revealed that core training improves basketball skill performance [36,38,40,43]. Several hypotheses couldexplain why core training differs from standard high-intensity strength training. This training method is based on the sports chain idea and does not split the sports chain. The abdominal, trunk, and hip muscles are the "heart" of the movement chain. They are crucial in the execution of upper and lower limb movements as well as the power transfer process [10]. Improved endurance of the core muscles of the waist and belly results inmore efficient transmission of lower limb strength.

Furthermore, for basketball skills, greater core stability enables athletes to efficiently convert strength to power, hence requiring less energy to accomplish skills [12]. When the core area's stability improves, players can better maintain their balance and posture [65]. Mayda et al. offered evidence that improving an athlete's body strength in a balanced manner facilitates the learning and practice of technical movements [66]. Thus, it is believed that a stable core could facilitate movement transfer during movement, thereby impactingbasketball players' skills [67]. In addition, basketball athletes must constantly change their body positions, maintain control and balance when using offensive skills, and strong core muscles provide a stable platform for athletes during fast, varying movements, improve the stability and control of athletes' dynamic and static postures, and increase the success rate of performing skills [68]. Consequently, core training most likely improves athletes' ability to control coordinated limb movements during a confrontation, more efficiently transfer upper and lower limb strength, and

reduce energy consumption for better skill display [69]. Nevertheless, additional experimental studies are required to confirm the findingsof this analysis.

## Limitations

This review has several limitations in addition to those stated in the screened articles. First, there is a dearth of researchon professional male basketball players. Second, research on other athletic and skill performances, such as endurance, flexibility, and basketball defensive skills, is lacking. Finally, the current review did not consider the uniqueness of basketball players' positions, as the experimental results may be affected when basketball players are tested on the same activity in different positions.

## Conclusion

Core training has the potential to improve athletic performance in terms of strength, sprinting, jumping, balancing, and agility, as well as skill performance in terms of shooting, passing, dribbling, rebounding, and stepping. However, current research on other athletic and skill performance,such as endurance, flexibility, and defensive skills, is lacking. Therefore, researchers can continue to explore these gaps in the field in order to help basketball players achieve better athletic and skill performance in the future.

## Practical application

Good core strength can act as a stable hub in the movement chain, helping athletes generate and transmit limb strength in a fierce confrontation. This strength helps the trunk quickly regain balance, improving basketball skills.This systematic review suggests combining static and dynamic core training. At the same time, coaches shouldnot ignore core training on unstable surfaces because this can achieve better results. Core training should include the plank, Superman support, and side bridge, as well as the Swiss ball as an instrument with an unstable surface. These exercises improvethe athletic and skill performances of basketball players. Therefore, it is suggested that basketball coaches should integrate core training into daily trainingsessions for at least four weeks, twice a week, to improve athletes' athletic and skill performance.

## Supporting information

**S1 Checklist. PRISMA 2020 checklist.**
(PDF)

## Author Contributions

**Conceptualization:** Shengyao Luo, Kim Geok Soh, Yanmei Zhao, Kim Lam Soh, He Sun, Xiuwen Zhai.

**Data curation:** Shengyao Luo, Yanmei Zhao, Kim Lam Soh, He Sun.

**Writing – original draft:** Shengyao Luo, Nasnoor Juzaily Mohd Nasiruddin, Luhong Ma.

**Writing – review & editing:** Shengyao Luo, Nasnoor Juzaily Mohd Nasiruddin, Luhong Ma.

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
