## [Decision Letter · Decision Letter 0]

20 Feb 2023

PONE-D-22-29930Effect of Core Training on Athletic and Skill Performance of Basketball Players: A Systematic ReviewPLOS ONE

Dear Dr. Zhao,

Thank you for submitting your manuscript to PLOS ONE. After careful consideration, we feel that it has merit but does not fully meet PLOS ONE’s publication criteria as it currently stands. Therefore, we invite you to submit a revised version of the manuscript that addresses the points raised during the review process.

We look forward to receiving your revised manuscript.

Kind regards,

Donald L. Hoover

Academic Editor

PLOS ONE

Journal Requirements:

Additional Editor Comments (if provided):

We believe this manuscript has promise, but issues raised by the peer reviewers must be addressed before it may meet the standards for publication. Thus, we encourage the authors to see reviewer reports, address the points raised, and resubmit in a timely fashion.

Reviewers' comments:

Reviewer's Responses to Questions

**Comments to the Author**

1. Is the manuscript technically sound, and do the data support the conclusions?

Reviewer #1: Yes

2. Has the statistical analysis been performed appropriately and rigorously? 

Reviewer #1: N/A

3. Have the authors made all data underlying the findings in their manuscript fully available?

Reviewer #1: Yes

4. Is the manuscript presented in an intelligible fashion and written in standard English?

Reviewer #1: Yes

5. Review Comments to the Author

Reviewer #1: Thank you for the opportunity to review this article, which bridges addresses the topic of core stability training and it’s potential impact upon sport-specific performance measures within the sport of basketball. This is a topic that has received relatively lesser attention in the scientific literature. Thus, it stands to reason that greater reasoned analysis of this topic may be helpful for sport-specific coaches, conditioning coaches, and sport scientists as to the potential value of this type of training as well as potential insights into how this type of specific training might reasonably be integrated into annual training calendars.

I see this manuscript as generally well-written and clearly laid out. The authors have done a nice job of laying out the need for such a study with the Introduction, clearly describing the Methods and Results, and tying the present findings to related studies within the Discussion.

In terms of content, my most substantive suggestion is that the authors could further address the potential ramifications of these findings in light of the “general motor abilities hypothesis” versus “specificity of motor ability hypothesis” that has existed for over half a century (the authors are encouraged to review Magill’s Motor Learning and Control, if they are not familiar with these theories, ad this author does a nice job of comparing and contrasting these specific theories: https://www.amazon.com/Motor-Learning-Control-Concepts-Applications/dp/1259823997). So, specifically, in reading and re-reading this manuscript three times in my review, I was intrigued by the basic premise of the present systematic review and how it may relate to these important theories in motor behavior. In this context, the authors broach the topic of “general strength training” versus “specific strength training” (e.g. core strength training) and the evidence that general strength training does not necessarily translate to improved performance on the basketball court, whereas the specific strength training does seem to carryover, in terms of sport-specific movements that help the athlete perform at a higher lever. The authors do a solid job of addressing current consensus on why the specifics of core training may help to facilitate the strength of the kinetic chain, but as I read lines 62-63, 88-89, 91-95, and 96-104 I am left wanting more information on how these concepts may relate to the “general motor abilities hypothesis” versus “specificity of motor ability hypothesis”. In sum, on this point most coaches, trainers, and sport scientists typically understand that all athletes cannot perform all forms of training, as there are not enough hours in the day nor the capacity to recover from some training in most athletes. So, finally, in terms of helping to shed light on the author’s primary purpose, weaving in this historical debate on the “general motor abilities hypothesis” versus “specificity of motor ability hypothesis”, they may be able to help readers better understand the “why” behind the favorability of the core strength training. This would further strengthen it’s potential contribution to the scientific literature. I would recommend that the authors address this theoretical content at the end of the paragraph which ends on line 108 (basically create a new paragraph addressing this content). They will then also need to revisit this conceptually within the Discussion, but again I think this would help to further strengthen this manuscript.

Otherwise, my suggestions for improving this manuscript are largely related to “cleaning up” some English-usage issues. The authors are likely writing in English as a second language, which likely has contributed to instances of odd language usage at times throughout the manuscript. Specific suggestions are listed below, but the authors are encouraged to further review their work for greater consistency in stylistic issues, as this will help to improve the overall quality.

In summary, the aim of this project has merit. I recommend the authors address the issues noted below prior to this manuscript appearing in this peer-reviewed journal. Thanks once again for the opportunity to review this paper.

Specific comments:

Line 31: Insert a comma after Google Scholar (e.g. “Google Scholar, were…”)

Line 38: Suggest revising from “Despite the lack of” to “Despite the relatively little evidence”

Line 50: remove “the”, and “basketball” should be expressed using lower case (not capitalized)

Line 62: Suggest revising to “as players do not demonstrate functional carryover as a result of this type of training.”

Line 64: Replace “affect” with “effect”

Lines 68-69: This sentence beginning “European and American….” does not thematically go with the rest of the paragraph, which focuses on core stability training. My suggestion is to revise this paragraph so that this sentence is moved to the previous paragraph, which actually addresses strength training.

Line 77: Suggest revising as follows: “… increasing strength transfer and coordinating muscle use and management during functional activities such as sport-specific skills.”

Line 105: Need a citation for this sentence.

Line 110: Suggest revising to “…but not performance with low specificity or functional carryover.”

Line 113: Need to be more specific here, describing no effect on what type of physical performance.

Line 127: Replace “by” with “until”

Lines 134-149: The first sentence in this passage is expressed in English past tense, as should be the case for this type of scientific writing. However, the second sentence then shifts to present tense, detracting from the authors’ clarity of expression. In short, this entire section should be revised so that it is expressed in past verb tense.

Line 139: Replace “genders” with “sexes”. This is the first instance of many in which the authors seem to use this terminology imprecisely. See the following sources for elaboration on these related, and often confused, terms:

https://cihr-irsc.gc.ca/e/48642.html

https://www.who.int/health-topics/gender#tab=tab_1

In sum, this study does not address socially-constructed gender roles but biological attributes associated with physical training, so it seems that “sex” should be used throughout this manuscript.

Line 158: Replace “gender” with “sex”

Line 164: Replace “ranges” with “ranged”

Line 170: Suggest revising to “….146 of which remained after duplicates were removed…”

Line 181: Suggest revising to “…the other studies did not report the characteristics of the participants.”

Line 181: Replace “gender” with “sex”

Line 199: Replace “includes” with “included”

Line 220: Replace “are” with “were”

Line 221: “plank” should not be capitalized (lower case)

Line 223: Capitalize “Superman”

Line 318: Suggest revising “waist and abdomen” to “trunk and hip”

Line 326: Replace “improves” with “improved”

Line 328: Replace “improves” with “improved”

Line 331: Replace “collect” with “absorb”

Line 335: Replace “guarantees” with “influences”

Line 339: Replace “extremists” with “extremities”

Lines 347-348: Suggest revising as follows: “Balance is regulated by the vestibular apparatus within the central nervous system as well as receptors in muscles, tendons, and joints and then coordinated within the somatosensory cortex with the aid of visual stimulus.”

Line 365: Suggest revising to “the abdominal, trunk, and hip muscles are the “heart” of the movement chain.”

Line 402: This is the first instance that you have capitalized “superman”. Technically, the superhero is known as “Superman”, so this likely needs to be capitalized throughout the manuscript simply for sake of consistency.

6. PLOS authors have the option to publish the peer review history of their article (what does this mean?). If published, this will include your full peer review and any attached files.

Reviewer #1: No

---

## [Author Response · Author response to Decision Letter 0]

14 Apr 2023

March 25, 2023

Donald L. Hoover

Academic Editor

PLOS ONE

Dear Prof. Donald L. Hoover

We would like to resubmit our revised manuscript for consideration of publication in PLOS ONE. The manuscript is entitled “Effect of Core Training on Athletic and Skill Performance of Basketball Players: A Systematic Review.” PONE-D-22-29930

The authors deeply appreciate the very detailed review of the manuscript and all the thoughtful comments that will undoubtedly help to improve it substantially. We have studied comments carefully and have made corrections which we hope meet with approval. Revised portions are made as track-change in the manuscript. The following is a point-by-point response to the reviewers’ comments. We hope that our manuscript can now be considered for publication in “PLOS ONE”. 

Comments from the editors and reviewers: 

Dear editors /referees many thanks for your constructive and valuable criticisms. Our responses are presented below and we are looking forward and ready to respond to any future comment (s)

Journal Requirements:

Response: Thank you for your insightful comment. We checked and modified the manuscript to ensure that it adhered to the journal’s guidelines.

Additional Editor Comments (if provided):

We believe this manuscript has promise, but issues raised by the peer reviewers must be addressed before it may meet the standards for publication. Thus, we encourage the authors to see reviewer reports, address the points raised, and resubmit in a timely fashion.

Response: Thank you very much for your encouraging comments and for recommending our manuscript. We are pleased to address all the reviewer’s comments as presented below. We hope our response meets your acceptance criteria. 

Reviewers' comments:

Reviewer's Responses to Questions

Comments to the Author

1. Is the manuscript technically sound, and do the data support the conclusions?

Reviewer #1: Yes

2. Has the statistical analysis been performed appropriately and rigorously?

Reviewer #1: N/A

3. Have the authors made all data underlying the findings in their manuscript fully available?

Reviewer #1: Yes

4. Is the manuscript presented in an intelligible fashion and written in standard English?

PLOS ONE does not copyedit accepted manuscripts, so the language in submitted articles must be clear, correct, and unambiguous. Any typographical or grammatical errors should be corrected at revision, so please note any specific err-ors here.

Reviewer #1: Yes

5. Review Comments to the Author

Reviewer #1: Thank you for the opportunity to review this article, which bridges addresses the topic of core stability training and it’s potential impact upon sport-specific performance measures within the sport of basketball. This is a topic that has received relatively lesser attention in the scientific literature. Thus, it stands to reason that greater reasoned analysis of this topic may be helpful for sport-specific coaches, conditioning coaches, and sport scientists as to the potential value of this type of training as well as potential insights into how this type of specific training might reasonably be integrated into annual training calendars.

I see this manuscript as generally well-written and clearly laid out. The authors have done a nice job of laying out the need for such a study with the Introduction, clearly describing the Methods and Results, and tying the present findings to related studies within the Discussion.

Response: Thank you very much for your encouraging comments and we are pleased to respond. Thank you for recommending our manuscript.

In terms of content, my most substantive suggestion is that the authors could further address the potential ramifications of these findings in light of the “general motor abilities hypothesis” versus “specificity of motor ability hypothesis” that has existed for over half a century (the authors are encouraged to review Magill’s Motor Learning and Control, if they are not familiar with these theories, ad this author does a nice job of comparing and contrasting these specific theories: https://www.amazon.com/Motor-Learning-Control-Concepts-Applications/dp/1259823997). So, specifically, in reading and re-reading this manuscript three times in my review, I was intrigued by the basic premise of the present systematic review and how it may relate to these important theories in motor behavior. In this context, the authors broach the topic of “general strength training” versus “specific strength training” (e.g. core strength training) and the evidence that general strength training does not necessarily translate to improved performance on the basketball court, whereas the specific strength training does seem to carryover, in terms of sport-specific movements that help the athlete perform at a higher lever. The authors do a solid job of addressing current consensus on why the specifics of core training may help to facilitate the strength of the kinetic chain, but as I read lines 62-63, 88-89, 91-95, and 96-104 I am left wanting more information on how these concepts may relate to the “general motor abilities hypothesis” versus “specificity of motor ability hypothesis”. In sum, on this point most coaches, trainers, and sport scientists typically understand that all athletes cannot perform all forms of training, as there are not enough hours in the day nor the capacity to recover from some training in most athletes. So, finally, in terms of helping to shed light on the author’s primary purpose, weaving in this historical debate on the “general motor abilities hypothesis” versus “specificity of motor ability hypothesis”, they may be able to help readers better understand the “why” behind the favorability of the core strength training. This would further strengthen it’s potential contribution to the scientific literature. I would recommend that the authors address this theoretical content at the end of the paragraph which ends on line 108 (basically create a new paragraph addressing this content). They will then also need to revisit this conceptually within the Discussion, but again I think this would help to further strengthen this manuscript.

Response: Thank you for your insightful comment. We have carefully considered your valuable suggestions and updated the introduction section with the following paragraph to demonstrate the historical debate on the “general motor abilities hypothesis” versus “specificity of motor ability hypothesis” (Lines: 110-124):

“There is a debate about the relationship of motor abilities; one viewpoint holds that motor abilities are highly related to one another (general motor abilities hypothesis), while the opposing view holds that they are relatively independent of one another (specific motor abilities theories). However, understanding the various points of view will aid in applying the concept of motor abilities to motor skill performance achievement [26]. The general motor abilities hypothesis has been around since the early twentieth century [27, 28]. It assumes that if a person is highly skilled in one motor skill, he or she will be or will become highly skilled in all motor skills. This prediction is based on the fact that there is only one general motor ability [26, 29]. The specificity of the motor abilities hypothesis is an alternative viewpoint that has received widespread support. Franklin Henry was widely credited with developing the specificity hypothesis in order to explain research findings that the general motor ability hypothesis could not explain [30]. According to this specificity viewpoint, individuals have numerous motor abilities that are relatively independent. This means that, for example, if a person demonstrated a high level of balancing ability, we couldn't predict how well that person would perform on a reaction time test [29].”

We also endorsed this concept in the discussion section to define the beneficial value of specific strength training over general strength training, as our study did for core strength training, in the following paragraphs (lines: 326-343): 

“There is sometimes confusion about whether we should use strength training in our training and what type of strength training we should use- when and how to apply it. General strength training focuses on the major muscle groups and can be beneficial during the off-season and even during the early base phase of training. It is increasing overall protective strength and will help protect specific joints from stresses that can occur in everyday life as well as during specific training sessions. Building muscle can also help with energy transfer- a mix of fast and slow twitch fibers can give you more energy uptake, which can help with endurance [45]. To build any type of muscle, however, we must overwork the muscle, which causes fatigue and necessitates recovery. Furthermore, most coaches, trainers, and sports scientists understand that most athletes do not have enough hours in the day or the capacity to recover from some training [46, 47]. 

As a result, specific strength training appears to be more beneficial because it can improve the range of motion within a sports activity and then get creative with the resistance of weights to build these muscles. It can stimulate specific muscle growth, increasing power, endurance, and speed without requiring excessive activity, capacity, time, or inducing fatigue [48]. This systematic review aimed to assess the impact of a specific strength training, i.e., core training, on basketball players' athletic and skill performance, as well as to offer recommendations for basketball coaches and potential directions for future research.”

Otherwise, my suggestions for improving this manuscript are largely related to “cleaning up” some English-usage issues. The authors are likely writing in English as a second language, which likely has contributed to instances of odd language usage at times throughout the manuscript. Specific suggestions are listed below, but the authors are encouraged to further review their work for greater consistency in stylistic issues, as this will help to improve the overall quality.

In summary, the aim of this project has merit. I recommend the authors address the issues noted below prior to this manuscript appearing in this peer-reviewed journal. Thanks once again for the opportunity to review this paper.

Response: Thank you very much for your encouraging comments and we reviewed the manuscript to correct any structural or grammatical errors.

Specific comments:

Line 31: Insert a comma after Google Scholar (e.g. “Google Scholar, were…”)

Line 38: Suggest revising from “Despite the lack of” to “Despite the relatively little evidence”

Line 50: remove “the”, and “basketball” should be expressed using lower case (not capitalized)

Line 62: Suggest revising to “as players do not demonstrate functional carryover as a result of this type of training.”

Line 64: Replace “affect” with “effect”

Lines 68-69: This sentence beginning “European and American….” does not thematically go with the rest of the paragraph, which focuses on core stability training. My suggestion is to revise this paragraph so that this sentence is moved to the previous paragraph, which actually addresses strength training.

Line 77: Suggest revising as follows: “… increasing strength transfer and coordinating muscle use and management during functional activities such as sport-specific skills.”

Line 105: Need a citation for this sentence.

Line 110: Suggest revising to “…but not performance with low specificity or functional carryover.”

Line 113: Need to be more specific here, describing no effect on what type of physical performance.

Line 127: Replace “by” with “until”

Lines 134-149: The first sentence in this passage is expressed in English past tense, as should be the case for this type of scientific writing. However, the second sentence then shifts to present tense, detracting from the authors’ clarity of expression. In short, this entire section should be revised so that it is expressed in past verb tense.

Line 139: Replace “genders” with “sexes”. This is the first instance of many in which the authors seem to use this terminology imprecisely. See the following sources for elaboration on these related, and often confused, terms:

https://cihr-irsc.gc.ca/e/48642.html

https://www.who.int/health-topics/gender#tab=tab_1

In sum, this study does not address socially-constructed gender roles but biological attributes associated with physical training, so it seems that “sex” should be used throughout this manuscript.

Line 158: Replace “gender” with “sex”

Line 164: Replace “ranges” with “ranged”

Line 170: Suggest revising to “….146 of which remained after duplicates were removed…”

Line 181: Suggest revising to “…the other studies did not report the characteristics of the participants.”

Line 181: Replace “gender” with “sex”

Line 199: Replace “includes” with “included”

Line 220: Replace “are” with “were”

Line 221: “plank” should not be capitalized (lower case)

Line 223: Capitalize “Superman”

Line 318: Suggest revising “waist and abdomen” to “trunk and hip”

Line 326: Replace “improves” with “improved”

Line 328: Replace “improves” with “improved”

Line 331: Replace “collect” with “absorb”

Line 335: Replace “guarantees” with “influences”

Line 339: Replace “extremists” with “extremities”

Lines 347-348: Suggest revising as follows: “Balance is regulated by the vestibular apparatus within the central nervous system as well as receptors in muscles, tendons, and joints and then coordinated within the somatosensory cortex with the aid of visual stimulus.”

Line 365: Suggest revising to “the abdominal, trunk, and hip muscles are the “heart” of the movement chain.”

Line 402: This is the first instance that you have capitalized “superman”. Technically, the superhero is known as “Superman”, so this likely needs to be capitalized throughout the manuscript simply for sake of consistency.

Response: Thank you for your careful review. We have carefully considered your valuable suggestions and corrected these errors throughout the manuscript. We appreciate your professional advice which helped us improve our manuscript.

Finally, we would like to express our gratitude to the editors and the reviewers for their time, efforts and their valuable comments.

Best regards;

Yanmei Zhao

cherryxf99@yxnu.edu.cn

Corresponding author

---

## [Decision Letter · Decision Letter 1]

29 May 2023

PONE-D-22-29930R1Effect of Core Training on Athletic and Skill Performance of Basketball Players: A Systematic ReviewPLOS ONE

Dear Dr. Zhao,

Thank you for submitting your manuscript to PLOS ONE. After careful consideration, we feel that it has merit but does not fully meet PLOS ONE’s publication criteria as it currently stands. Therefore, we invite you to submit a revised version of the manuscript that addresses the points raised during the review process.

We look forward to receiving your revised manuscript.

Kind regards,

Donald L. Hoover

Academic Editor

PLOS ONE

Journal Requirements:

Additional Editor Comments:

The authors have done a nice job of addressing reviewer suggestions and revising this manuscript. The re-review created a few more minor suggested edits. The authors are encouraged to make these revisions in a timely manner and then resubmit.

Reviewers' comments:

Reviewer's Responses to Questions

**Comments to the Author**

1. If the authors have adequately addressed your comments raised in a previous round of review and you feel that this manuscript is now acceptable for publication, you may indicate that here to bypass the “Comments to the Author” section, enter your conflict of interest statement in the “Confidential to Editor” section, and submit your "Accept" recommendation.

Reviewer #1: (No Response)

2. Is the manuscript technically sound, and do the data support the conclusions?

Reviewer #1: Yes

3. Has the statistical analysis been performed appropriately and rigorously? 

Reviewer #1: Yes

4. Have the authors made all data underlying the findings in their manuscript fully available?

Reviewer #1: (No Response)

5. Is the manuscript presented in an intelligible fashion and written in standard English?

Reviewer #1: Yes

6. Review Comments to the Author

Reviewer #1: Thank you for the opportunity to re-review this article, which bridges addresses the topic of core stability training and its potential impact upon sport-specific performance measures within the sport of basketball. As I noted in my original review, this topic has received little attention in the scientific literature. Additional information on this topic may be helpful for sport-specific coaches, conditioning coaches, and sport scientists. In sum, I believe the authors have adequately addressed my suggestion for better explaining the “why” behind the mechanisms that may contribute to this functional carryover seen with core training.

As noted in my original review, I see this manuscript as generally well-written and clearly laid out. In the revision, the authors have done a fine job of integrating my original suggestion regarding the inclusion of motor behavior theory and how it may reasonably illustrate the conceptual rationale for how this type of core training may carry over to increased functional performance on sport-specific skills within the sport of basketball. The value of this cannot be overstated, given the goal of all coaches to make links between the training they engage athletes in and the carryover to on-court performance.

Otherwise, my suggestions for improving this manuscript are largely related to “cleaning up” a few additional English-usage issues. In re-reading this manuscript, I have found a few junctures where I believe the word usage can be improved, so I’ve made some specific friendly edits toward the goal of contributing to the overall “readability” of the manuscript.

In summary, the aim of this project has merit. I recommend the authors address the issues noted below prior to this manuscript appearing in this peer-reviewed journal. Thanks once again for the opportunity to review this paper.

Specific comments:

Lines 59-60: Suggest revising to the following: “European and American experts began to widely implement strength training principles in a variety of sports in the late 1990s.”

Line 116: Need a citation here, at the end of this sentence. The Magill textbook will work fine here (29).

Line 127: Suggest adding the following sentence to the end of this paragraph: “In sum, the specificity hypothesis is likely helpful in understanding in what ways core strength training may contribute to improved sport-specific performance.”

Line 138: Suggest a minor revision as follows: “This systematic review aims to elucidate the impact of core training as a function of the above-noted specificity hypothesis on basketball players’ physical and skill performance.” Restating in this manner will help to better tie in the new content the authors have included regarding the general vs specificity concept, thus helping to better elucidate why they have elected to do this study.

Line 134: Need to correct this typo. I assume it’s referring to VO2 max but not clear as written.

Line 166: Revise to the following: “…addressing the impact of at least one core strength intervention on…”

Line 257: Replace “are” with “were”

Line 341: Suggest revising to: “There is a lack of consensus on whether an athlete should use strength training within a sport-specific training program, what type of strength training should the athlete use, when to use, and how to apply it.” Citations are needed for this sentence; number 7 likely works here but others are needed too.

Line 345: Need a citation here, at the end of the sentence. Also, suggest revising to “General strength training can provide overall protective strength ….”

Line 346: Suggest revising to: “Adding additional muscle can also help with energy transfer, as a mix of fast and slow twitch fibers can provide more energy uptake, which can help athletes with endurance [45]”.

Line 348: Suggest revising to: “To build any type of muscle, however, the muscle must be overloaded, which causes fatigue and necessitates recovery.”

Line 350: Suggest revising to: “Furthermore, most coaches, trainers, and sport scientists understand that most athletes do not possess unlimited hours each day to train nor do they have limitless capacity to recover from training and competition, so they must be highly strategic in what they include in training plans [46,47].”

Line 352: Suggest revising to: “As a result, highly specific strength training appears to be most beneficial to athletes, as it can improve the execution of sports activities, particularly as external resistance may be added to make the training task more difficult. Specific strength training can stimulate focused muscle growth, increasing further power, endurance, and speed of movement without requiring the excessive time commitments or undue fatigue that generalized strength training programs may be likely to generate [48]”

7. PLOS authors have the option to publish the peer review history of their article (what does this mean?). If published, this will include your full peer review and any attached files.

Reviewer #1: No

---

## [Author Response · Author response to Decision Letter 1]

3 Jun 2023

June 1, 2023

Donald L. Hoover

Academic Editor

PLOS ONE

Dear Prof. Donald L. Hoover

We would like to resubmit our revised manuscript for consideration of publication in PLOS ONE. The manuscript is entitled “Effect of Core Training on Athletic and Skill Performance of Basketball Players: A Systematic Review.” PONE-D-22-29930

The authors deeply appreciate the very detailed review of the manuscript and all the thoughtful comments that will undoubtedly help to improve it substantially. We have studied comments carefully and have made corrections which we hope meet with approval. Revised portions are made as track-change in the manuscript. The following is a point-by-point response to the reviewers’ comments. We hope that our manuscript can now be considered for publication in “PLOS ONE”. 

Comments from the editors and reviewers: 

Dear editors /referees many thanks for your constructive and valuable criticisms. Our responses are presented below and we are looking forward and ready to respond to any future comment (s)

Journal Requirements:

Journal Requirements:

Response: Thank you for your insightful comment. We checked the reference list to ensure that it is complete and correct.

Additional Editor Comments (if provided):

The authors have done a nice job of addressing reviewer suggestions and revising this manuscript. The re-review created a few more minor suggested edits. The authors are encouraged to make these revisions in a timely manner and then resubmit.

Response: Thank you very much for your encouraging comments and for recommending our manuscript. We are pleased to address all the reviewer’s comments as presented below. We hope our response meets your acceptance criteria. 

Reviewers' comments:

Reviewer's Responses to Questions

Comments to the Author

1. If the authors have adequately addressed your comments raised in a previous round of review and you feel that this manuscript is now acceptable for publication, you may indicate that here to bypass the “Comments to the Author” section, enter your conflict of interest statement in the “Confidential to Editor” section, and submit your "Accept" recommendation.

Reviewer #1: (No Response)________________________________________

2. Is the manuscript technically sound, and do the data support the conclusions?

Reviewer #1: Yes

3. Has the statistical analysis been performed appropriately and rigorously?

Reviewer #1: Yes

4. Have the authors made all data underlying the findings in their manuscript fully available?

Reviewer #1: (No Response)

5. Is the manuscript presented in an intelligible fashion and written in standard English?

Reviewer #1: Yes

6. Review Comments to the Author

Reviewer #1: Thank you for the opportunity to re-review this article, which bridges addresses the topic of core stability training and its potential impact upon sport-specific performance measures within the sport of basketball. As I noted in my original review, this topic has received little attention in the scientific literature. Additional information on this topic may be helpful for sport-specific coaches, conditioning coaches, and sport scientists. In sum, I believe the authors have adequately addressed my suggestion for better explaining the “why” behind the mechanisms that may contribute to this functional carryover seen with core training.

As noted in my original review, I see this manuscript as generally well-written and clearly laid out. In the revision, the authors have done a fine job of integrating my original suggestion regarding the inclusion of motor behavior theory and how it may reasonably illustrate the conceptual rationale for how this type of core training may carry over to increased functional performance on sport-specific skills within the sport of basketball. The value of this cannot be overstated, given the goal of all coaches to make links between the training they engage athletes in and the carryover to on-court performance.

Otherwise, my suggestions for improving this manuscript are largely related to “cleaning up” a few additional English-usage issues. In re-reading this manuscript, I have found a few junctures where I believe the word usage can be improved, so I’ve made some specific friendly edits toward the goal of contributing to the overall “readability” of the manuscript.

In summary, the aim of this project has merit. I recommend the authors address the issues noted below prior to this manuscript appearing in this peer-reviewed journal. Thanks once again for the opportunity to review this paper.

Response: Thank you very much for your encouraging comments and we are pleased to respond. Thank you for recommending our manuscript.

Specific comments:

Lines 59-60: Suggest revising to the following: “European and American experts began to widely implement strength training principles in a variety of sports in the late 1990s.”

Response: Thank you for your careful review. We have carefully considered your valuable suggestions and revised the sentence (lines: 59-61).

Line 116: Need a citation here, at the end of this sentence. The Magill textbook will work fine here (29).

Response: Thank you for your insightful comment. We included the suggested reference (line 116).

Line 127: Suggest adding the following sentence to the end of this paragraph: “In sum, the specificity hypothesis is likely helpful in understanding in what ways core strength training may contribute to improved sport-specific performance.”

Response: Thank you for your careful review. We have carefully considered your valuable suggestions and revised the sentence (lines: 127-129).

Line 138: Suggest a minor revision as follows: “This systematic review aims to elucidate the impact of core training as a function of the above-noted specificity hypothesis on basketball players’ physical and skill performance.” Restating in this manner will help to better tie in the new content the authors have included regarding the general vs specificity concept, thus helping to better elucidate why they have elected to do this study.

Response: Thank you for your careful review. We have carefully considered your valuable suggestions and revised the sentence (lines: 139-141).

Line 134: Need to correct this typo. I assume it’s referring to VO2 max but not clear as written.

Response: Thank you for your careful review. We have corrected this typo (line 136).

Line 166: Revise to the following: “…addressing the impact of at least one core strength intervention on…”

Response: Thank you for your careful review. We have carefully considered your valuable suggestions and revised the sentence (lines: 163-164).

Line 257: Replace “are” with “were”

Response: Thank you for your careful review. We have carefully considered your valuable suggestions and revised this word (line 251).

Line 341: Suggest revising to: “There is a lack of consensus on whether an athlete should use strength training within a sport-specific training program, what type of strength training should the athlete use, when to use, and how to apply it.” Citations are needed for this sentence; number 7 likely works here but others are needed too.

Response: Thank you for your careful review. We have carefully considered your valuable suggestions and revised the sentence (lines: 334-336). We also included the recommended citation and two other ones.

Gabbett TJ. The training-injury prevention paradox: should athletes be training smarter and harder?. Br J Sports Med. 2016;50(5):273-280. doi:10.1136/bjsports-2015-095788

Granacher U, Lesinski M, Büsch D, et al. Effects of Resistance Training in Youth Athletes on Muscular Fitness and Athletic Performance: A Conceptual Model for Long-Term Athlete Development. Front Physiol. 2016;7:164. Published 2016 May 9. doi:10.3389/fphys.2016.00164

Line 345: Need a citation here, at the end of the sentence. Also, suggest revising to “General strength training can provide overall protective strength ….”

Response: Thank you for your careful review. We have carefully considered your valuable suggestions and revised the sentence (lines: 338-339). We also included a citation as recommended.

Hughes DC, Ellefsen S, Baar K. Adaptations to Endurance and Strength Training. Cold Spring Harb Perspect Med. 2018 Jun 1;8(6):a029769. doi: 10.1101/cshperspect.a029769. PMID: 28490537; PMCID: PMC5983157.

Line 346: Suggest revising to: “Adding additional muscle can also help with energy transfer, as a mix of fast and slow twitch fibers can provide more energy uptake, which can help athletes with endurance [45]”.

Response: Thank you for your careful review. We have carefully considered your valuable suggestions and revised the sentence (lines: 342-344).

Line 348: Suggest revising to: “To build any type of muscle, however, the muscle must be overloaded, which causes fatigue and necessitates recovery.”

Response: Thank you for your careful review. We have carefully considered your valuable suggestions and revised the sentence (lines: 346-347).

Line 350: Suggest revising to: “Furthermore, most coaches, trainers, and sport scientists understand that most athletes do not possess unlimited hours each day to train nor do they have limitless capacity to recover from training and competition, so they must be highly strategic in what they include in training plans [46,47].”

Response: Thank you for your careful review. We have carefully considered your valuable suggestions and revised the sentence (lines: 349-352).

Line 352: Suggest revising to: “As a result, highly specific strength training appears to be most beneficial to athletes, as it can improve the execution of sports activities, particularly as external resistance may be added to make the training task more difficult. Specific strength training can stimulate focused muscle growth, increasing further power, endurance, and speed of movement without requiring the excessive time commitments or undue fatigue that generalized strength training programs may be likely to generate [48]”

Response: Thank you for your careful review. We have carefully considered your valuable suggestions and revised the sentence (lines: 354-359).

Finally, we would like to express our gratitude to the editors and the reviewers for their time, efforts and their valuable comments which helped us improve our manuscript.

Best regards;

Yanmei Zhao

cherryxf99@yxnu.edu.cn

Corresponding author

---

## [Editor Report · Decision Letter 2]

5 Jun 2023

Effect of Core Training on Athletic and Skill Performance of Basketball Players: A Systematic Review

PONE-D-22-29930R2

Dear Dr. Zhao,

We’re pleased to inform you that your manuscript has been judged scientifically suitable for publication and will be formally accepted for publication once it meets all outstanding technical requirements.

Kind regards,

Donald L. Hoover

Academic Editor

PLOS ONE

Additional Editor Comments (optional):

The authors have adequately addressed each specific issue of constructive criticism and/or suggestion. My recommendation is to accept this manuscript.
---

## [Editor Report · Acceptance letter]

12 Jun 2023

PONE-D-22-29930R2 

Effect of Core Training on Athletic and Skill Performance of Basketball Players: A Systematic Review 

Dear Dr. Zhao:

I'm pleased to inform you that your manuscript has been deemed suitable for publication in PLOS ONE. Congratulations! Your manuscript is now with our production department. 

Kind regards, 

on behalf of

Dr. Donald L. Hoover 

Academic Editor

PLOS ONE